# A phase division-based multi-segment foot model for estimating dynamic foot arch stiffness during walking

Chenhao Liu[1], Jingang Yi[2], Long He[3], Yijun Zhang[4], Tao Liu[1]*

**1** State Key Laboratory of Fluid Power and Mechatronic Systems, School of Mechanical Engineering, Zhejiang University, Hangzhou, Zhejiang, China, **2** Department of Mechanical and Aerospace Engineering, Rutgers University, Piscataway, New Jersey, United States of America, **3** Zhiyuan Research Institute, Hangzhou, China, **4** Department of Orthopedics, The First Affiliated Hospital, Zhejiang University School of Medicine, Hangzhou, China

* liutao@zju.edu.cn

**Data availability statement:** Data cannot be shared publicly because of legal and ethical concerns. Data are available from the Medical Ethics Committee of the School of Biomedical

## Abstract

The arch of the human foot plays a significant role in bearing weight and keeping gait balance. Previous studies mainly focus on the foot arch stiffness at the static or quasi-dynamic state of a particular foot shape. The variation of the linear arch stiffness across the entire walking gait has rarely been reported. This work presents a phase division-based multi-segment foot model that considers plantar aponeurosis's tension force for calculating the dynamics of the medial longitudinal arch. Kinematics and ground reaction forces of 10 healthy young adults during walking are recorded and analyzed. The stiffness changes of the foot arch throughout the stance phase are calculated. The experimental results show that the foot arch experiences a stiff-compliant-stiff-compliant transition during a single stance phase, including an extremely low stiffness during the plantar contact phase. By comparing the foot arch stiffness results with those from previous studies, the accuracy of the proposed model is indirectly validated. This study presents a new approach to explore the variation of the linear stiffness of the foot arch across the entire stance phase during walking. The proposed multi-segment foot model provides a new method for solving foot dynamics that can be used for wearable sensing and assistive design and applications.

## Introduction

The human foot possesses a complex biomechanical structure that contains numerous bones, joints, muscles, ligaments, nerves, and vessels. Among these components, the foot arch plays a vital role in bearing the weight of the human body, absorbing the ground impact [1], distributing the pressure on the sole [2], and maintaining balance [3]. In great apes and other primates, no medial longitudinal arch (MLA) structure similar to humans has been found [4]. The stiffness of the arch is constantly changing during the stance phase. In clinical thought, the foot arch is compliant at ground contact but stiff during propulsion [5], playing a critical factor in human bipedal walking and running [4].

Engineering and Instrument Science, Zhejiang University (contact via e-mail: huacow@zju.edu.cn) for researchers who meet the criteria for access to confidential data.

**Funding:** This work was supported in part by the National Natural Science Foundation of China Grant No. U1913601, 52175033 and U21A20120. The funders had no role in study design, data collection and analysis, decision to publish, or preparation of the manuscript.

**Competing interests:** The authors have declared that no competing interests exist.

Estimation of the variation of the forces on the foot arch during walking is of great significance to understanding the energetics of gait. The foot arch's compression and recoil over the stance phase are considered a vital contribution to the economy of walking locomotion [6]. The foot arch absorbs energy from the ground impact when compressed and then returns energy for propulsion when recoiled [7], with the dynamics of the foot arch changing constantly. Stearne et al. (2016) [8] demonstrated that during running, the compression/recoil of the foot arch helps reduce energy cost. The passive elastic work of the foot arch can largely explain the energy savings it provides, which would otherwise need to be performed by active muscles.

The stiffness of lower limb joints such as the hip [9] and knee [10] has been estimated and exploited to design robotic exoskeletons and prosthetics. Traditional foot arch support devices [11–13] can ensure the standard height of the wearer's foot arch by adding rigid protrusions under the arch. Nevertheless, such rigid, passive support does not account for changes in arch stiffness and could limit the compression and recoil of the foot arch, increasing the energy cost of running [8]. Understanding the changing characteristics of arch stiffness during the stance phase is crucial for designing foot orthoses and insoles, diagnosing and treating patients with flat feet, and designing and controlling other wearable assistive devices for the lower limb.

Although many studies [14,15] have measured and discussed the dynamic foot arch stiffness during walking, they all calculated the rotational arch moment using inverse dynamics based on ground reaction force (GRF). The joint moment-angle relationship is typically how dynamic arch stiffness is measured. Kern et al. (2019) [15] calculated midtarsal joint quasi-stiffness during walking with added mass and summarized the average stiffness during dorsiflexion and plantarflexion. However, linear arch stiffness is more consistent with the physical description of the 'arch' structure, as demonstrated by other studies measuring arch stiffness in a static foot state [7,16]. Measuring the variation of linear arch stiffness during walking or running in vivo is challenging since we cannot directly measure the load on the foot arch when moving.

Many foot models have been proposed [17–20] accounting for the foot arch and the mechanism influencing its stiffness. Farris et al. (2019) [18] found minimal support for the role of intrinsic foot muscles in supporting the arch in early and mid-stance during walking and running. Farris et al. (2020) [21] also demonstrated that active muscle contraction, rather than the passive windlass mechanism [22], primarily contributed to foot stiffness during bipedal walking propulsion. Additionally, active contraction of the ankle dorsiflexor muscles provides tension in the plantar fascia. At the same time, intrinsic foot muscles contribute to tension development in the plantar region, offering rigidity to the foot arch. Therefore, this study primarily focuses on the tension force of plantar aponeurosis (PA) as a critical factor influencing foot arch stiffness. In this study, the height change of the foot arch is used to describe the deformation, as defined by Venkadesan et al. (2020) [16], and the force on the foot arch will be calculated by a novel multi-segment foot model in different stance phases.

In this study, we present a multi-segment foot model for calculating the dynamics of the MLA based on PA's tension force. The kinematic data of critical points in the model are measured to obtain the displacement of the height of the foot arch. The GRF during the stance phase is recorded to estimate the force acting on the foot arch based on the model. Subsequently, the foot arch linear stiffness during the stance phase is estimated and analyzed. Comparing the foot arch stiffness results with those from previous studies provides indirect validation of the accuracy of the proposed model. The contribution of this work lies in proposing a novel foot model that divides the stance phase into three phases and conducting force

analysis on different segments of the foot during gait phases. Based on this model, the variation of the arch stiffness during the walking stance phase is estimated. The proposed phase division-based multi-segment foot model provides a new method for solving foot dynamics.

## Materials and methods

### Participants

Ten young, healthy subjects (Seven males and three females, age: $24 \pm 2$ years, height: $1.72 \pm 0.05$ meters, mass: $67 \pm 11$ kg) were informed and gave their written consent and participated in the study. This study was approved by the Medical Ethics Committee of the School of Biomedical Engineering and Instrument Science, Zhejiang University (Project identification code: 2021-39). The recruitment period for the subjects in this study was from December 1, 2021 to October 31, 2022. All subjects had neutral foot arches and had no current injuries or past medical history in the lower limbs. Individuals with flat feet, high arches, or other abnormal arches were excluded from the final group of participants.

### Foot model

A phase division-based multi-segment foot model considering PA's tension force is proposed, as shown in Fig 1(a). The model is a 2D-foot representation determined by the plane of the MLA, and it includes three essential parts related to human motion: the ankle, the MLA, and the metatarsophalangeal joint (MTPJ). The weight of the foot is small compared with the load on the foot or GRFs, and therefore, the gravity on the foot is neglected. The whole foot is divided into forefoot and rearfoot, with navicular bone as the boundary [23]. Mechanically, the forefoot and rearfoot are firmly connected at the navicular bone, and their interaction includes unknown forces and moments (both vectors) in the 2D plane. The stance phase of gait can be divided into three phases: heel contact phase (from the heel strike to the foot flat), plantar contact phase, and push-off phase.

In this model, we consider both the active contraction of intrinsic foot muscles and the passive stretching force of the PA. The active contraction of intrinsic muscles can elevate the arch, while the passive stretching of the PA can lower it. However, their commonality is that both exert forces on the arch structure. Since the direction of the intrinsic muscle contraction force cannot be directly obtained, we assume that the direction of the forces from both factors aligns with that of the PA. Therefore, the 'tension force of the PA' mentioned later refers to the combined effect of active muscle contraction and passive stretching forces.

Fig 1(b) illustrates the free-body diagrams of the foot in these phases, indicating the foot's dynamics during the stance phase. During the heel contact phase, only considering the rearfoot since it is in contact with the ground, the model's equations of motion are obtained from Newton's second law. The generalized coordinates of this model are set at the position and orientation in the vertical plane, and the equations of motion are given by

$$
\begin{bmatrix} m_R & 0 & 0 \\ 0 & m_R & 0 \\ 0 & 0 & J_R \end{bmatrix} \begin{bmatrix} \ddot{x}_R \\ \ddot{y}_R \\ \ddot{\theta}_R \end{bmatrix} = \begin{bmatrix} F_{f1} + F_{PA} - F_{BW}\cos\theta_B - F_{Ax} \\ F_{GR1} - F_{BW}\sin\theta_B - F_{Ay} \\ F_{BW}l_B + F_{Ay}x_A - F_{Ax}y_A - M_A \end{bmatrix} \tag{1}
$$

where $(x_R, y_R)$ is the position of the center of mass of the rearfoot, $(x_A, y_A)$ is the coordinate of arch break (navicular tuberosity), $\theta_R$ is the rotation angle of the rearfoot relative to the origin $O$, $\theta_B$ is the direction in which body weight $F_{BW}$ exerts a force on the rearfoot, $l_B$ is the moment arm, $m_R$ is the mass of the rearfoot, $J_R$ is the moment of inertia of rearfoot about the

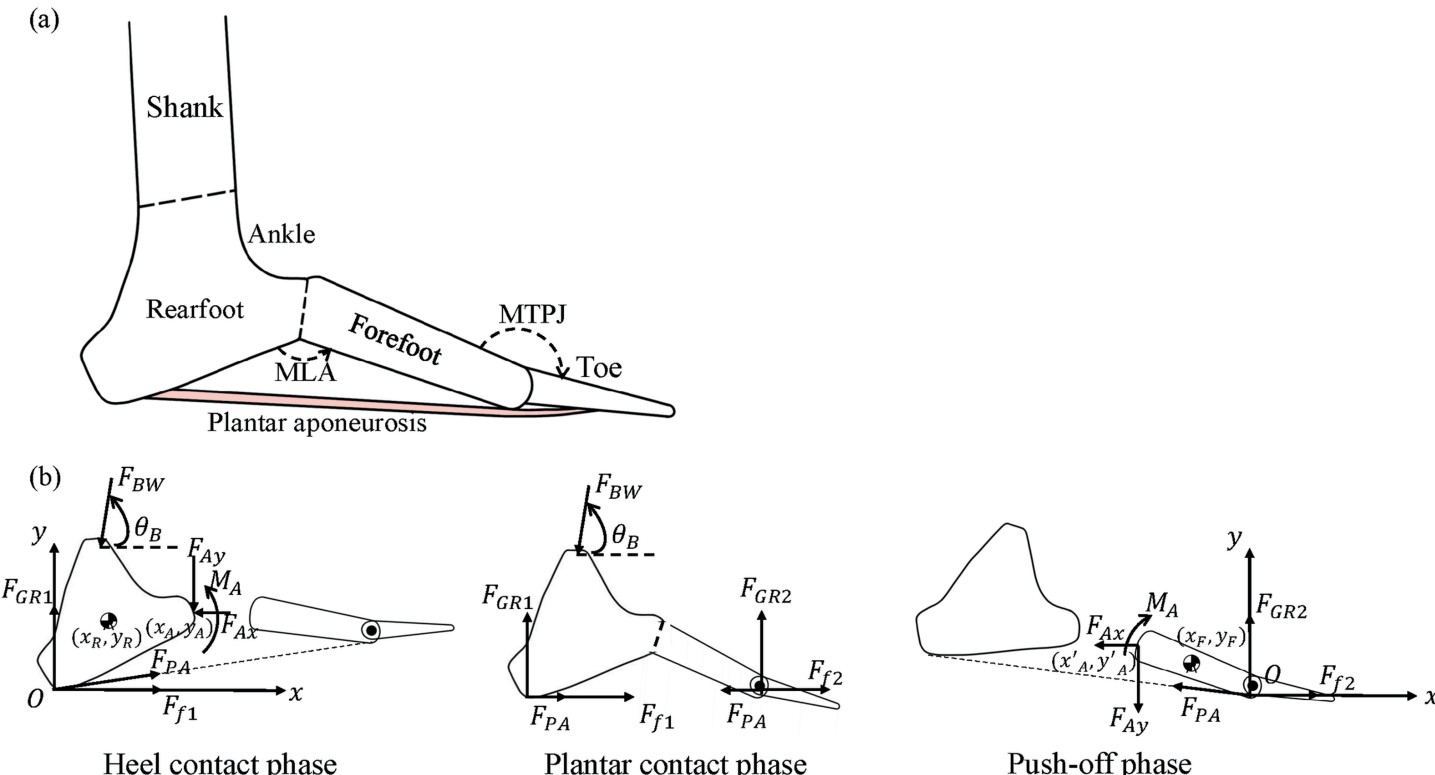

**Fig 1. (a) The multi-segment foot model consisted of a shank, rearfoot, forefoot, and toes.** The model includes three essential parts related to human motion: ankle, arch, and metatarsophalangeal joint, considering the PA's tension force. (b) The free-body-diagrams of the foot during the heel contact phase, plantar contact phase, and push-off phase.

origin, $F_{f1}$ and $F_{GR1}$ are horizontal and vertical reaction force between the ground and the sole of the rearfoot, and $\begin{bmatrix} F_{Ax} & F_{Ay} & M_A \end{bmatrix}$ is a set of internal force of arch exerted by the firm connection of forefoot and rearfoot; see Fig 1. Moreover, according to the basic minimal biped walking model [24,25], the ground reaction force ($F_{GR1}$) is equal to the load exerted by the body on the foot ($F_{BW}$). $F_{PA}$ is the tension force of PA, whose variations during the stance phase have been investigated through cadaveric studies [26], human experiments [27], and finite element analysis [28]. The estimation of $F_{PA}$ extends beyond the scope of this study, and we directly adopt the tension force variation curve from Chen et al. (2015) [28]. Finally, noting that $F_{Ax}$, $F_{Ay}$, and $M_A$ are the only remaining unknowns, three equations for planar motion are obtained so that we have

$$\begin{cases} F_{Ax} = F_{f1} + F_{PA} - F_{GR1}\cos\theta_B - m_R\ddot{x}_R \\ F_{Ay} = (1 - \sin\theta_B)F_{GR1} - m_R\ddot{y}_R \\ M_A = F_{GR1}l_B + F_{Ay}x_A - F_{Ax}y_A - J_R\ddot{\theta}_R \end{cases} \qquad (2)$$

The load on the foot arch is the internal force of the foot caused by the plantar pressure and body weight. Therefore, we define the load on the arch:

$$F_{AL} = \sqrt{F_{Ax}^2 + F_{Ay}^2} \qquad (3)$$

During the plantar contact phase, the foot is considered as a whole. The whole foot is dynamically balanced, and the inertial force is negligible. Therefore, we take the static or quasi-dynamic foot state processing method [14,16]. Considering that the load applied to the arch structure is equivalent to the downward force exerted by the body on the foot, we obtain

$$F_{AL} = F_{BW} = \sqrt{F_{GR1}^2 + F_{f1}^2} + \sqrt{F_{GR2}^2 + F_{f2}^2} \tag{4}$$

where the right-hand side of Eq (4) represents the overall GRFs.

Similarly, during the push-off phase, the origin of the coordinates of the model is set at the first metatarsophalangeal joint. The ground-contacted forefoot is considered, and the equations of motion are written as:

$$\begin{bmatrix} m_F & 0 & 0 \\ 0 & m_F & 0 \\ 0 & 0 & J_F \end{bmatrix} \begin{bmatrix} \ddot{x}_F \\ \ddot{y}_F \\ \ddot{\theta}_F \end{bmatrix} = \begin{bmatrix} F_{f2} - F_{PA} - F_{Ax} \\ F_{GR2} - F_{Ay} \\ -F_{Ay}x'_A - F_{Ax}y'_A + M_A \end{bmatrix} \tag{5}$$

where $(x_F, y_F)$ is the position of the center of mass of the forefoot, $(x'_A, y'_A)$ is the coordinate of arch relative to the new origin, $\theta_F$ is the rotation angle of the forefoot relative to the origin, $m_F$ is the mass of the forefoot, $J_F$ is the moment of inertia of forefoot about the origin, and the $F_{f2}$ and $F_{GR2}$ are horizontal and vertical reaction force between the ground and the sole of the forefoot. From Eq (5), we obtain

$$\begin{cases} F_{Ax} = F_{f2} + F_{PA} - m_F\ddot{x}_F \\ F_{Ay} = F_{GR2} - m_F\ddot{y}_F \\ M_A = F_{Ay}x'_A + F_{Ax}y'_A + J_F\ddot{\theta}_F \end{cases} \tag{6}$$

and the load on the foot arch here is also defined as Eq (3).

The stiffness of foot arch $k$ is defined as the changes of load applied to the foot $\Delta F_{AL}$ versus the height change of the foot arch $\Delta h$, that is,

$$k = \frac{\Delta F_{AL}}{\Delta h} \tag{7}$$

and this applies to the heel contact phase, plantar contact phase, and push-off phase. We calculate the arch stiffness continuously by combining Eq (2), Eq (3), and Eq (7) for the heel contact phase, Eq (4) and Eq (7) for the plantar contact phase, and Eq (3), Eq (6), and Eq (7) for the push-off phase.

It must be noted that the stiffness calculation is the change in force at the arch versus the height of the arch. The change in force should be considered only along the direction in which height is measured. In calculation, the force at the arch we used was the component of $\Delta F_{AL}$ in the direction of the $\Delta h$. All kinematic data and GRFs will be measured experimentally. The definition and measurement of the arch height $h$ will be discussed in the next session.

## Kinematic and kinetic data of foot

Foot kinematic data and GRFs were recorded to solve the inverse dynamics of the foot arch throughout the stance phase. Three-dimensional motion data were captured at 100 Hz using a 10-infrared camera motion capture system. Meanwhile, GRFs were collected synchronously at 1000 Hz using two 6-dimension force plates. The layout of the experimental site is shown

in Fig 2(a). The force plates were placed in a staggered position to ensure that a complete stance phase of each foot was measured. A total of 12 retro-reflective markers were attached to landmarks on both feet of each participant, which included a posterior aspect of the calcaneus, first metatarsal head, tuberosity of the navicular bone, lateral side of the Hallux [29], medial ankle, and end of the calf, as shown in Fig 2(b). It should be noted that the marker at the medial ankle and end of the calf was used to determine $\theta_B$. All the marked points were used not only to obtain the kinematics of the foot but also to obtain its geometric data. All the subjects were told to walk on the 6-meter-long track [30] at a comfortable pace and cadence for five repetitions, each beginning and ending with a period of statically standing. The static foot arch height of each subject was recorded and calculated during the standing. Each subject was given time and opportunity to acclimate in advance. They were guaranteed to complete at least three full gait cycles during each walk.

All kinematic and GRF data were filtered offline using moving average filters. To apply our proposed 2D-foot dynamics model, the 3D position of all markers (in the world coordinate) and GRF data were transformed to the x-y plane where the MLA was located by a spatial mapping algorithm. The x direction of the vertical plane was the direction of the posterior aspect of the calcaneus pointing to the first metatarsal head, and the y direction was the vertical upward direction, which was the z direction in the original world coordinate system. Our analysis is predicated on the center of pressure of GRF, which is identified with one of the three segments of the foot.

As noted in Fig 2(b), the navicular height was defined as the height of the arch of each participant. After projecting the coordinates of all markers onto the proposed 2D plane, the foot arch height was calculated as the distance from the navicular bone marker to the line formed

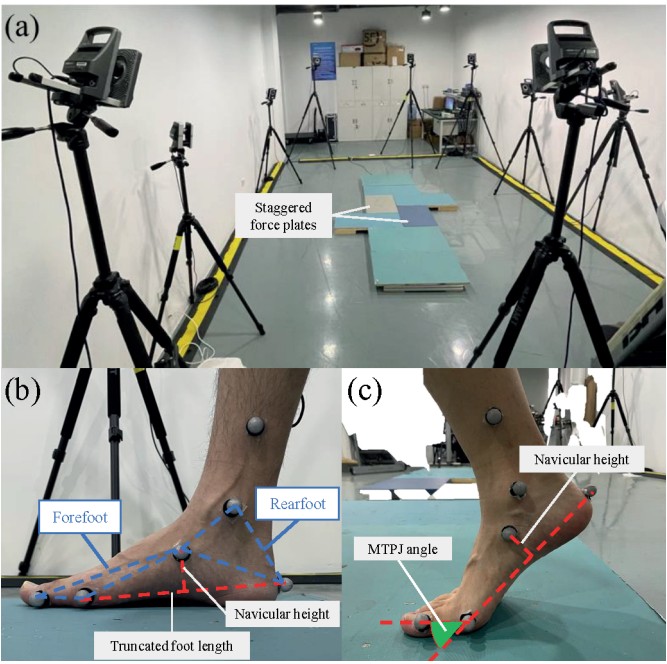

**Fig 2. (a) The layout of the experimental site with a 10-infrared camera motion capture system and two force plates.** (b) The location of the six markers at one foot and the definition of arch height. (c) The definition of MTPJ angle.

by the markers on the calcaneus and the first metatarsal head. The forefoot plane was defined by three markers at the Hallux, first metatarsal head, and navicular bone, whereas the rear-foot plane was defined by the markers at the navicular bone, ankle, and posterior aspect of the calcaneus. And these two planes are merged together at the navicular bone. For plotting and comparison, the truncated foot length (the distance from the posterior aspect of the heel to the first metatarsal head) was used for normalization to obtain the normalized navicular height truncation (NNHt) as defined by Murley et al. (2009) [31]. Moreover, the MTPJ angle or dorsiflexion of the toe was defined through two vectors, which were spanned as shown in Fig 2(c). The stance phase was defined with a 30N vertical ground reaction force threshold. The heel contact phase, plantar contact phase, and push-off phase are distinguished by the height of the posterior aspect of the calcaneus and first metatarsal head with a 20 mm height threshold, considering the size of the markers. All kinematic and kinetic data were synchronized, and time was normalized to the stance phase duration for plotting and visual inspection.

## Model accuracy validation protocol

It is crucial to validate the accuracy of the proposed multi-segment foot model. The model established in this paper aims to calculate, or estimate, the load on the foot's arch. However, measuring the load on the foot arch during walking is highly challenging, and there is currently no standard protocol to validate the accuracy of foot biomechanics models. In this study, we validate the accuracy of our model by comparing the calculated dynamic stiffness of the foot arch with results from existing studies, including quasi-dynamic, dynamic, linear stiffness, and rotational stiffness studies. This comparison serves as the model validation method, similar to the approach used by Bruening et al. (2012) [23]. The parameters selected for comparison include: the trend of changes in dynamic stiffness and the range of results. Moreover, we also calculate the coefficient of determination ($R^2$) between the force-displacement (or moment-angle) curves of the foot arch in previous studies and our work. The comparison of the trend in dynamic stiffness variation and the $R^2$ results represent the similarity in the variation of foot arch dynamic stiffness between this study and previous studies. The comparison of stiffness result ranges reflects the validity of the results in this study.

Data from the force-displacement (or moment-angle) curves of the foot arch in previous studies were extracted using image data extraction software. These data were then compared with the results of this study. After aligning the data points through interpolation and scaling the values, the coefficient of determination ($R^2$) was calculated using standard statistical formulas to assess the similarity in the variation of foot arch stiffness. Quasi-dynamic and static foot measurement studies will be excluded from the comparison of the trend in dynamic stiffness variation and $R^2$ calculation. Studies with stiffness results in different units from this study will be excluded from the comparison of result ranges.

## Results

The subdivision of the stance phase was determined, and the changes in foot arch height, load, and stiffness throughout the stance phase were calculated, as presented in Fig 3. Fig 3(a) shows the changes in the height of markers at the posterior calcaneus and first metatarsal head. According to their changes, we divided one standing phase into the heel contact phase, plantar contact phase, and push-off phase, which were indicated by a color area. The change in arch height during the stance phase is shown in Fig 3(b), and the calculated force at the foot arch is plotted in Fig 3(c). For all subjects, the height of the foot arch continued to decline after heel contact, and there was a period of decline at the beginning of the push-off phase,

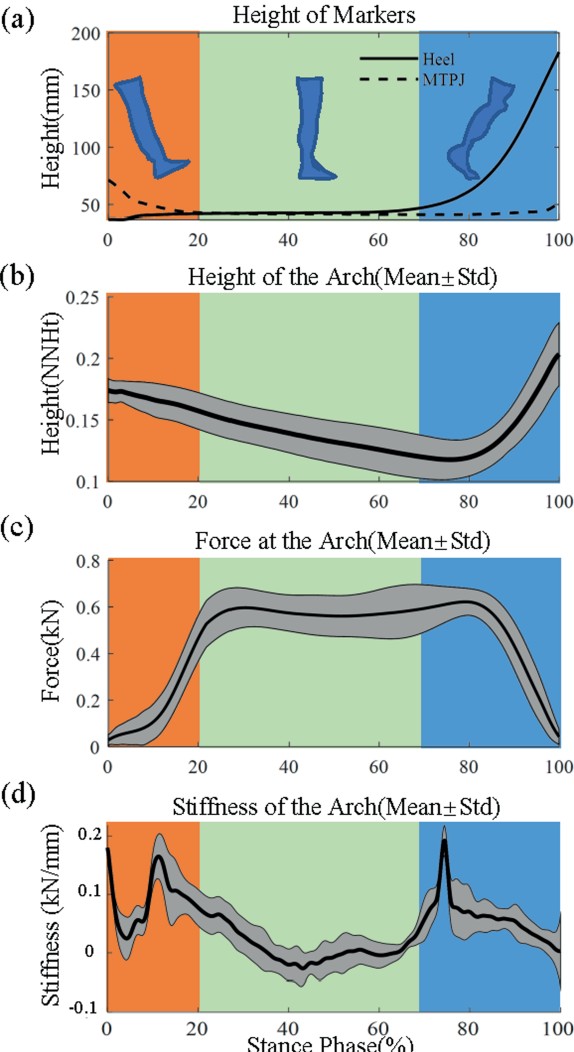

**Fig 3. (a) The changes in the height of markers at the posterior calcaneus and first metatarsal head (mean of ten subjects) and the subdivision of the stance phase.** (b) The change in arch height during the stance phase (mean±standard deviation). (c) The force at the arch during the stance phase (mean±standard deviation). (d) The variation of the foot arch stiffness across the stance phase (mean±standard deviation).

and then it began to rise rapidly. Additionally, Fig 3(d) presents the variation of the arch stiffness across the stance phase. As the foot contacted the ground and the arch descended, the arch stiffness gradually decreased at the middle of the plantar contact phase. After entering the push-off stage, the stiffness of the arch raised sharply. Then, as the foot moved forward, the arch stiffness gradually decreased until the foot left the ground.

Fig 4 shows an example of the load on the foot arch versus the downward displacement of the foot arch of one subject. The direction of the gait progression is marked in the figure, and the average arch stiffness was estimated by linearly fitting the curve at the beginning of the heel contact phase, the plantar contact phase, and the end of the push-off phase. The slopes of the three fitting lines were 0.124 kN/mm (Root Mean Square Error (RMSE)=1.589%) for the

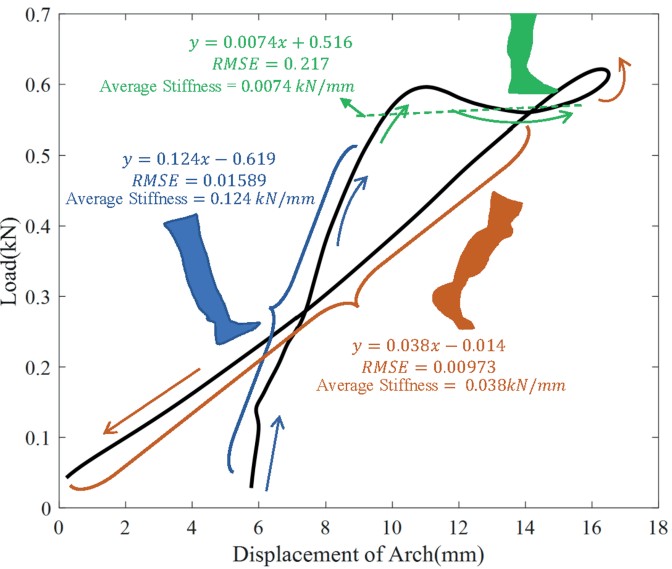

**Fig 4. An example of the load on the foot arch versus the downward displacement of the foot arch of a subject, and the fitted slopes during early heel contact, plantar contact, and late propulsion.**

beginning of the heel contact phase, 0.0074 kN/mm (RMSE=21.7%) for the the plantar contact phase, and 0.038 kN/mm (RMSE=0.973%) for the end of the push-off phase. Moreover, Table 1 lists the calculated values of ten subjects' average arch stiffness in these three periods, along with their sex, mass, truncated foot length, normalized static arch height and maximum arch displacement. The mean and standard deviation of the parameters were also calculated and noted. It can be seen that every subject's foot arch stiffness turned extremely low during the plantar contact phase, and the average stiffness was 0.0069 kN/mm. Furthermore, it is clear from the table that the average arch stiffness in early heel contact was approximately three times that during late propulsion.

Table 2 presents the results of the model accuracy validation, comparing the trend and range of dynamic stiffness variation calculated by the model proposed in this study with those from previous studies, as well as calculating the coefficient of determination ($R^2$) between the force-displacement (or moment-angle) curves of the foot arch to characterize the similarity.

As shown in Table 2, for studies with stiffness results in the same units as those defined in this study, the range of foot arch stiffness is generally between 0 and 0.2 kN/mm, which is consistent with the results of this study. For studies with different units but also calculating dynamic stiffness variations, the stiffness variation trends are largely consistent when viewed from the force/displacement curves. These studies all exhibit stiffness during the heel contact phase, compliance during the plantar contact phase, a sudden increase in stiffness, and compliance during the push-off phase. The $R^2$ values from curve comparisons indicate that these results all have $R^2$ values above 0.6, with the highest reaching 0.88. These comparative results support the validity of the model proposed in this study.

## Discussion

This work presented a new multi-segment foot model to explore the human body's force on the foot arch and the variation of the foot arch linear stiffness during walking gait. The load

**Table 1. Details of subjects' foot and the fitted average stiffness during early heel contact, plantar contact, and late propulsion.**

| Subjects | Sex | Mass(kg) | Truncated foot length (mm) | Static arch height (NNHt) | Maximum arch displacement (NNHt) | $k_{HC}$ (kN/mm) | $k_{PC}$ (kN/mm) | $k_{PO}$ (kN/mm) |
|---|---|---|---|---|---|---|---|---|
| 1 | F | 52 | 188.7 | 0.164 | 0.106 | 0.073 | 0.0081 | 0.028 |
| 2 | F | 58 | 184.1 | 0.161 | 0.090 | 0.124 | 0.0074 | 0.038 |
| 3 | F | 61 | 179.6 | 0.156 | 0.095 | 0.127 | 0.0067 | 0.029 |
| 4 | M | 56 | 204.1 | 0.175 | 0.072 | 0.121 | 0.0065 | 0.038 |
| 5 | M | 57 | 194.3 | 0.165 | 0.081 | 0.079 | 0.0071 | 0.059 |
| 6 | M | 72 | 197.9 | 0.162 | 0.092 | 0.130 | 0.0061 | 0.041 |
| 7 | M | 69 | 177.0 | 0.169 | 0.072 | 0.166 | 0.0066 | 0.053 |
| 8 | M | 84 | 206.3 | 0.155 | 0.071 | 0.149 | 0.0072 | 0.051 |
| 9 | M | 75 | 212.6 | 0.158 | 0.072 | 0.181 | 0.0066 | 0.043 |
| 10 | M | 82 | 214.8 | 0.154 | 0.071 | 0.133 | 0.0065 | 0.035 |
| | **Mean** | **66.6** | **197.0** | **0.162** | **0.082** | **0.128** | **0.0069** | **0.042** |
| | **STD** | **11.4** | **13.5** | **0.007** | **0.013** | **0.034** | $\mathbf{5.8 \times 10^{-4}}$ | **0.010** |

$k_{HC}$: average arch stiffness during the beginning of heel contact phase.
$k_{PC}$: average arch stiffness during the plantar contact phase.
$k_{PO}$: average arch stiffness during the end of push-off phase.
F: female, M: male.
STD: Standard deviation.

**Table 2. Comparison of the trend and range of dynamic stiffness variation calculated by the model proposed in this study with those from previous studies and the coefficient of determination ($R^2$) between the force-displacement (or moment-angle) curves.**

| Foot model | Trend of arch stiffness | Range of arch stiffness variation | Coefficient of determination ($R^2$) |
|---|---|---|---|
| This study | stiff-compliant-stiff-compliant | 0 – 0.188 kN/mm | – |
| Welte et al. (2021) [7] | – | 0.042 – 0.211 kN/mm | – |
| Sanchis-Sales et al. (2019) [14] | stiff-compliant-stiff-compliant | – | 0.61 |
| Kern et al. (2019) [15] | stiff-compliant-stiff-compliant | – | 0.88 |
| Venkadesan et al. (2020) [16] | – | 0 – 0.375 kN/mm | – |
| Davis et al. (2022) [17] | compliant-stiff-compliant | – | 0.66 |
| Kondo et al. (2021) [32] | stiff-compliant-stiff-compliant | – | 0.74 |

– Not found or not applicable because it is a quasi-dynamic/static foot measurement or the units are different.

on the foot arch was estimated by dividing the stance phase into three stages through a simplified model of the foot dynamics. Our results indicated that the foot arch underwent a stiff-compliant-stiff-compliant transition during a single stance phase. The foot arch is rigid when the foot touches the ground, and then the foot arch height gradually decreases. What stands out in the results is a period of extremely low arch stiffness before the foot moves forward. Then, a rapid increase in arch stiffness is present during the push-off phase, and the foot arch stiffness gradually decreases until the foot leaves the ground.

Table 3 presents a comparison between the proposed foot model and other foot models in the literature. In terms of definition, many studies quantify arch deformation using inter-joint angles [17,18] or forefoot-rearfoot angles [14,32]. Similar to this work, some studies adopt navicular height as the metric [7,16]. Regarding dynamics, almost all existing research directly utilizes the GRF or its derived moments to represent the force applied to the foot or arch. This study presents the phase division-based multi-segment model that not only calculates the force on the arch but also circumvents the need to measure various internal forces that are difficult to quantify directly, such as the plantar flexor force and intrinsic muscle forces during

**Table 3. Comparison between the proposed foot model and previous models for the human foot arch.**

| Foot model | Force | Displacement | Mechanism | Utility in *vivo* |
|---|---|---|---|---|
| This study | Calculated by model | Navicular height of the arch | Tension force of PA | Dynamic foot arch stiffness |
| Sanchis-Sales et al. (2019) [14] | Calculated moments | Midtarsal joint angle | Windlass mechanism | Dynamic foot arch stiffness |
| Kern et al. (2019) [15] | Calculated moments | Multi-Joints angle | MTPJ dorsiflexion | Quasi-stiffness |
| Davis et al. (2022) [17] | GRF | Cal-Met$^{(*)}$+ angle | Tension force of PA and muscles | MTPJ stiffness |
| Kondo et al. (2021) [32] | GRF | R-F$^{(+)}$ angle | Windlass mechanism | Fixed value of arch stiffness |

+ R-F: rearfoot segment with respect to the forefoot segment.
* Cal-Met: calcaneus and metatarsal segments.

the push-off phase. Additionally, our model incorporates the active tension on the PA instead of solely relying on the passive windlass mechanism, which is controversial in terms of its contribution to arch stiffness. Ultimately, the proposed model enables continuous estimation of dynamic foot arch stiffness throughout the stance phase.

Although the changes in linear arch stiffness during the stance phase have rarely been reported, we can still compare the arch stiffness results with those in the existing studies, as shown in Table 2. Venkadesan et al. (2020) [16] measured the stiffness of the cadaver foot in a static state, and the definition of arch stiffness is the same as in this paper. The stiffness of the foot arch varies between 0 – 0.375 kN/mm when placed flat on the ground. Kondo et al. (2021) [32] used a fixed value to represent the overall stiffness of the foot arch in the stance phase, but the arch stiffness was defined as the ground reaction force versus the angle between the forefoot and rearfoot. Nevertheless, the extremely low stiffness in the plantar contact phase was also found in their results. In this study, a reduction in the force at the arch was observed during a specific period of the plantar contact phase while the foot arch height continued to decrease. This phenomenon has been documented in other studies that directly utilize GRF to assess foot arch loading [18,32]. However, in studies employing moments to model external forces acting on the foot [33], the moment consistently increased, thereby indicating the absence of negative stiffness. This discrepancy arises from differences in definition and essentially highlights the highly compliant nature of the foot arch during the plantar contact phase.

The proposed multi-segment foot model provides a new method for solving foot dynamics. However, there are some limitations to this study. First, we used the tension force of PA to represent the combined effects of the passive windlass mechanism and active muscle forces and hypothesized that the direction of the forces from both factors aligns with that of the PA. Second, we directly utilized the PA's tension force from existing literature for calculation. Additionally, we used 2-D representation of the foot arch instead of 3-D, sacrificing the accuracy. Finally, due to the lack of practical methods for measuring foot arch loads, the model accuracy validation in this study is incomplete. This limitation arises because the proposed model is specifically designed to calculate foot arch loads and cannot provide other mechanical quantities.

Prospects involve considering the windlass mechanism, calf muscles, ankle muscles and ligaments, and intrinsic foot muscles in 3-D perspective to further improve the proposed model. In terms of segmentation methods for multi-segment foot models, future work includes considering the addition of the midfoot to better align the model with the anatomy of the foot, thereby improving accuracy. Prospects also includes the potential to enhance the

model by adding complexity, allowing for the calculation of other easily measurable physical quantities, thereby improving the accuracy validation method. The varying characteristics of the arch stiffness across the entire stance phase can serve as an inspiration and reference for foot orthoses and wearable assistive devices [34,35]. Regarding design and assisting control, foot orthoses, insoles, or assistive devices should consider the foot's dynamic stiffness. Completely passive and rigid foot orthotics and insoles may not be appropriate. The ideal assistive device should have the ability of dynamic regulation to assist when the foot stiffens and become flexible when the foot arch reduces in height [36].

## Conclusion

This paper presents a new approach to investigate the variation of foot arch linear stiffness throughout the stance phase of walking. We established a phase division-based multi-segment foot model to estimate the load on the arch during three phases by measuring ground reaction forces and solving inverse dynamics. Based on the kinetic and displacement data, the foot arch's stiffness variation was estimated throughout the whole stance phase of walking. Extremely low stiffness during the plantar contact phase and an increasingly high stiffness during early propulsion were found. The foot arch experienced a stiff-compliant-stiff-compliant transition across the whole stance phase. The proposed multi-segment foot model can be further applied to other dynamics analysis and walking studies of the foot. This work also provided inspiration and reference for designing and controlling foot orthotics and intelligent assistive devices for the lower limb.

## Author contributions

**Conceptualization:** Chenhao Liu, Jingang Yi, Long He, Yijun Zhang, Tao Liu.

**Data curation:** Yijun Zhang, Tao Liu.

**Formal analysis:** Jingang Yi.

**Funding acquisition:** Tao Liu.

**Investigation:** Chenhao Liu, Long He, Tao Liu.

**Methodology:** Chenhao Liu, Yijun Zhang.

**Supervision:** Tao Liu.

**Validation:** Chenhao Liu, Long He.

**Writing – original draft:** Chenhao Liu, Jingang Yi, Tao Liu.

**Writing – review & editing:** Chenhao Liu, Jingang Yi, Yijun Zhang, Tao Liu.

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
