## [Decision Letter · Decision Letter 0]

29 Dec 2024

PONE-D-24-51122A Phase Division-Based Multi-Segment Foot Model for Estimating Dynamic Foot Arch Stiffness During WalkingPLOS ONE

Dear Dr. Liu,

Thank you for submitting your manuscript to PLOS ONE. After careful consideration, we feel that it has merit but does not fully meet PLOS ONE’s publication criteria as it currently stands. Therefore, we invite you to submit a revised version of the manuscript that addresses the points raised during the review process.

We look forward to receiving your revised manuscript.

Kind regards,

Fei Yan

Academic Editor

PLOS ONE

“This work was supported in part by the National Natural Science Foundation of China Grant No. U1913601, 52175033 and U21A20120.”

4. For studies involving third-party data, we encourage authors to share any data specific to their analyses that they can legally distribute. PLOS recognizes, however, that authors may be using third-party data they do not have the rights to share. When third-party data cannot be publicly shared, authors must provide all information necessary for interested researchers to apply to gain access to the data. (https://journals.plos.org/plosone/s/data-availability#loc-acceptable-data-access-restrictions)

Reviewers' comments:

Reviewer's Responses to Questions

**Comments to the Author**

1. Is the manuscript technically sound, and do the data support the conclusions?

Reviewer #1: Yes

Reviewer #2: Partly

2. Has the statistical analysis been performed appropriately and rigorously? 

Reviewer #1: Yes

Reviewer #2: Yes

3. Have the authors made all data underlying the findings in their manuscript fully available?

Reviewer #1: Yes

Reviewer #2: Yes

4. Is the manuscript presented in an intelligible fashion and written in standard English?

Reviewer #1: Yes

Reviewer #2: Yes

5. Review Comments to the Author

Reviewer #1: In this study, the authors presented a phase division-based multi-segment foot model that considered plantar aponeurosis's tension force for calculating the dynamics of the medial longitudinal arch. The results showed that the foot arch experienced a stiff compliant-stiff-compliant transition during a single stance phase, including an extremely low stiffness during the plantar contact phase. The method is interesting and the results are reliable, however there are few questions need to be clarified before publication.

1. in the foot model part, the authors claimed that “The whole foot is divided into forefoot and rearfoot, with navicular bone as the boundary.”. We usually divide foot into three parts: forefoot, midfoot and rearfoot. Why did follow the anatomy of foot? Can you remodel the foot and compare the discrepancy?

2. in the table 1, did you normalized the change of arch height with arch length? Why not?

3. In figure 4 description text, “The slopes of the three fitting lines were 0.124 kN/mm (Root Mean Square Error (RMSE)=1.589%) for the beginning of the heel contact phase, 0.0074 kN/mm (RMSE=0.211%) for the the plantar contact phase, and 0.038 kN/mm (RMSE=0.973%) for the end of the push-off phase”. While in the figure picture, it was marked with the 0.0074kN/mm(RMSE=0.217). Is this a typo?

Reviewer #2: This paper proposes a multi-segment foot model to explore the forces on the foot arch and the variation of the foot arch linear stiffness during walking gait. The study has certain value, but there are still some issues that need to be addressed.

1. How is the foot arch height calculated in the three phases of the stance phase?

2. Validating the accuracy of the proposed model is crucial. This section needs to be added.

3. What is the difference between calculating foot arch variation using foot arch angle changes versus using foot arch height changes?

4. In Table 2, the paper compares with other studies. In addition to differences in theoretical calculations, what are the differences in computational accuracy and results?

6. PLOS authors have the option to publish the peer review history of their article (what does this mean?). If published, this will include your full peer review and any attached files.

Reviewer #1: No

Reviewer #2: No

---

## [Author Response · Author response to Decision Letter 1]

4 Feb 2025

Response to the Academic Editor

Comments to the Author:

Thank you for submitting your manuscript to PLOS ONE. After careful consideration, we feel that it has merit but does not fully meet PLOS ONE’s publication criteria as it currently stands. Therefore, we invite you to submit a revised version of the manuscript that addresses the points raised during the review process.

We appreciate the Academic Editor’s help for inviting experts to review the paper. The reviewers have offered invaluable feedback, providing a wealth of insightful queries and suggestions. In response to reviewer feedback, we have enhanced our manuscript by addressing the following: we conducted a more thorough literature review, added the rationale behind the modeling approach, clarified the result data, defined the method for calculating foot arch height, included a model accuracy validation section, addressed the limitations of the current study, and provided an outlook for future research. The technical merit of the research has been clarified and enriched as a result of this review process, consequently enhancing the manuscript's readability.

Response to Reviewer # 1

In this study, the authors presented a phase division-based multi-segment foot model that considered plantar aponeurosis's tension force for calculating the dynamics of the medial longitudinal arch. The results showed that the foot arch experienced a stiff compliant-stiff-compliant transition during a single stance phase, including an extremely low stiffness during the plantar contact phase. The method is interesting and the results are reliable, however there are few questions need to be clarified before publication.

Thanks for your careful review and suggestions for improving the manuscript. Revisions have been made according to your suggestions.

1. in the foot model part, the authors claimed that “The whole foot is divided into forefoot and rearfoot, with navicular bone as the boundary.”. We usually divide foot into three parts: forefoot, midfoot and rearfoot. Why did follow the anatomy of foot? Can you remodel the foot and compare the discrepancy?

Response 1: We appreciate your careful review and professional comments. We deeply agree that the division of the foot into forefoot, midfoot, and rearfoot is the more commonly used anatomical model in both clinical and anatomical contexts. However, forefoot-rearfoot model with the navicular as a boundary are also used in many studies.

The reason for establishing the arch model by dividing it into forefoot and rearfoot mainly has two aspects. First, in this paper, the purpose of proposing this model is to facilitate the calculation of the loads acting on the foot arch. By dividing the foot into forefoot and rearfoot, with the arch as the dividing line, it is more effective in achieving our goal. Second, we refer to the multi-segment foot model proposed by D. A. Bruening et al. 2012 [1] (doi: 10.1016/j.gaitpost.2011.10.363), which states: “The model consists of a Shank (tibia and fibula) and three foot segments: (1) Hindfoot (calcaneus and talus), (2) Forefoot (navicular, cuboid, cuneiforms, and metatarsals), and (3) Hallux (proximal and distal phalanges). While additional segments may be useful in future work (e.g. medial/lateral and/or midfoot/forefoot segmentation), they currently present hurdles in force measurement and repeatability”. We have already added this citation in the Foot Model section: “The whole foot is divided into forefoot and rearfoot, with navicular bone as the boundary [23].”

[1]. Bruening DA, Cooney KM, Buczek FL. Analysis of a kinetic multi-segment foot model. Part I: Model repeatability and kinematic validity. Gait & Posture. 2012;35:529–534.

At the same time, the models used in the numerous studies we cited that attempt to estimate the dynamic stiffness of the foot arch use the forefoot and rearfoot segmentation method. Here we list some literature in our citations:

[2]. Sanchis-Sales E, Sancho-Bru JL, Roda-Sales A, Pascual-Huerta J. Variability of the Dynamic Stiffness of Foot Joints: Effect of Gait Speed. Journal of the American Podiatric Medical Association. 2019;109:291–298.

[3]. Kern AM, Papachatzis N, Patterson JM, Bruening DA, Takahashi KZ. Ankle and midtarsal joint quasi-stiffness during walking with added mass. PeerJ. 2019;7:e7487.

[4]. Davis DJ, Challis JH. Foot arch rigidity in walking: In vivo evidence for the contribution of metatarsophalangeal joint dorsiflexion. PLoS ONE. 2022;17:e0274141.

[5]. Bruening DA, Cooney KM, Buczek FL. Analysis of a kinetic multi-segment foot model part II: Kinetics and clinical implications. Gait & Posture. 2012;35:535–540.

Certainly, many studies have incorporated the midfoot into multi-segment foot models to calculate the foot stiffness [6, 7], underscoring its important role. However, we respectfully acknowledge that it would be beyond the scope and focus of our current study to re-model the foot using the three-segment approach and compare the differences between the two methods. We have, however, included a discussion on the potential for future work using the three-segment anatomical model in the last paragraph of the Discussion section: “In terms of segmentation methods for multi-segment foot models, future work includes considering the addition of the midfoot to better align the model with the anatomy of the foot, thereby improving accuracy.”

[6]. Farris DJ, Kelly LA, Cresswell AG, Lichtwark GA. The functional importance of human foot muscles for bipedal locomotion. Proceedings of the National Academy of Sciences. 2019;116:1645–1650.

[7] Zhu, S., Jenkyn, T. Development of a clinically useful multi-segment kinetic foot model. J Foot Ankle Res 16, 86 (2023). https://doi.org/10.1186/s13047-023-00686-0

2. in the table 1, did you normalized the change of arch height with arch length? Why not?

Response 2: Thank you for your comments. There must be some misunderstanding. We are confident that both the Static Arch Height and Maximum Arch Displacement in Table 1 have been normalized using the Truncated Foot Length, as indicated by the units labeled "NNHt" in the table. This aligns with the explanation provided in Section Kinematic and Kinetic Data of Foot, which states: “For plotting and comparison, the truncated foot length (the distance from the posterior aspect of the heel to the first metatarsal head) was used for normalization to obtain the normalized navicular height truncation (NNHt) as defined by Murley et al. [31]”

To improve readability and assist with understanding, we have slightly revised the introductory statement in the second paragraph of the Results section, which corresponds to Table 1. The revised statement now reads: “Moreover, Table 1 lists the calculated values of ten subjects’ average arch stiffness in these three periods, along with their sex, mass, truncated foot length, normalized static arch height and maximum arch displacement.”

3. In figure 4 description text, “The slopes of the three fitting lines were 0.124 kN/mm (Root Mean Square Error (RMSE)=1.589%) for the beginning of the heel contact phase, 0.0074 kN/mm (RMSE=0.211%) for the the plantar contact phase, and 0.038 kN/mm (RMSE=0.973%) for the end of the push-off phase”. While in the figure picture, it was marked with the 0.0074kN/mm(RMSE=0.217). Is this a typo?

Response 3: Thanks for your careful review and comments. This is a typo. We apologize for our mistake. We have made changes to the corresponding Figure 4 description text: “The slopes of the three fitting lines were 0.124 kN/mm (Root Mean Square Error (RMSE)=1.589%) for the beginning of the heel contact phase, 0.0074 kN/mm (RMSE=21.7%) for the plantar contact phase, and 0.038 kN/mm (RMSE=0.973%) for the end of the push-off phase”.

Response to Reviewer # 2

This paper proposes a multi-segment foot model to explore the forces on the foot arch and the variation of the foot arch linear stiffness during walking gait. The study has certain value, but there are still some issues that need to be addressed.

Thank you for your thorough review of this manuscript and for the time and effort you have invested in reviewing. This manuscript has been fully revised according to your professional suggestions.

1. How is the foot arch height calculated in the three phases of the stance phase?

Response 1: We appreciate your careful review and valuable comments. In fact, the method for calculating foot arch height remains consistent across the three phases of the stance phase. As stated in Section Kinematic and Kinetic Data of Foot, on page 5, in the last paragraph: “As noted in Fig. 2(b), the navicular height was defined as the height of the arch of each participant.”

Fig.2 (a) The layout of the experimental site with a 10-infrared camera motion capture system and two force plates. (b) The location of the six markers at one foot and the definition of arch height. (c) The definition of MTPJ angle.

To enhance readability and facilitate understanding, we have added further clarification after that text: “After projecting the coordinates of all markers onto the proposed 2D plane, the foot arch height was calculated as the distance from the navicular bone marker to the line formed by the markers on the calcaneus and the first metatarsal head.”

2. Validating the accuracy of the proposed model is crucial. This section needs to be added.

Response 2: Thank you for your critical comments. Like many other studies on foot arch stiffness [1-5], the model presented in this paper is designed to calculate, or estimate, the load on the foot arch. It is important to clarify that, similar to the reason why these references lack model accuracy validation, the absence of such validation in this study is not due to a lack of awareness of its importance. As mentioned in the paper, measuring the load on the foot arch in vivo during walking is extremely challenging. Additionally, using modeling software such as OpenSim for indirect calculations to validate the accuracy of our proposed model would be tangential to the main focus of this study.

This explains why it is difficult to validate our model through direct experimental measurements or software simulations. Therefore, we can only validate the model’s accuracy as best as possible by comparing the dynamic stiffness of the foot arch calculated by our model with results from existing studies. This serves as the accuracy validation section for our model. This approach is also the method used by D. A. Bruening et al. (2012, doi: 10.1016/j.gaitpost.2011.10.363) [6] in their study to validate model accuracy. The parameters selected for comparison include: the trend of changes in dynamic stiffness and the range of results. Moreover, we also calculate the coefficient of determination (R²) between the force-displacement (or moment-angle) curves of the foot arch in previous studies and our work. The comparison of the trend in dynamic stiffness variation and the R² results represent the similarity in the variation of foot arch dynamic stiffness between this study and previous studies. The comparison of stiffness result ranges reflects the validity of the results in this study.

In the revised manuscript, we have made the following improvements:

We added this clarification in Abstract: “By comparing the foot arch stiffness results with those from previous studies, the accuracy of the proposed model is indirectly validated.”

We also added this sentence in the last paragraph of Section Introduction: “Comparing the foot arch stiffness results with those from previous studies provides indirect validation of the accuracy of the proposed model.”

Most importantly, we added a new Subsection Model Accuracy Validation Protocol in the end of Section Materials and methods:

“It is crucial to validate the accuracy of the proposed multi-segment foot model. The model established in this paper aims to calculate, or estimate, the load on the foot's arch. However, measuring the load on the foot arch during walking is highly challenging, and there is currently no standard protocol to validate the accuracy of foot biomechanics models. In this study, we validate the accuracy of our model by comparing the calculated dynamic stiffness of the foot arch with results from existing studies, including quasi-dynamic, dynamic, linear stiffness, and rotational stiffness studies. This comparison serves as the model validation method, similar to the approach used by Bruening et al. (2012) [23]. The parameters selected for comparison include: the trend of changes in dynamic stiffness and the range of results. Moreover, we also calculate the coefficient of determination (R²) between the force-displacement (or moment-angle) curves of the foot arch in previous studies and our work. The comparison of the trend in dynamic stiffness variation and the R² results represent the similarity in the variation of foot arch dynamic stiffness between this study and previous studies. The comparison of stiffness result ranges reflects the validity of the results in this study.

Data from the force-displacement (or moment-angle) curves of the foot arch in previous studies were extracted using image data extraction software. These data were then compared with the results of this study. After aligning the data points through interpolation and scaling the values, the coefficient of determination (R²) was calculated using standard statistical formulas to assess the similarity in the variation of foot arch stiffness. Quasi-dynamic and static foot measurement studies will be excluded from the comparison of the trend in dynamic stiffness variation and R² calculation. Studies with stiffness results in different units from this study will be excluded from the comparison of result ranges.”

In the Section Results, we also added a Table and a paragraph:

“Table 2 presents the results of the model accuracy validation, comparing the trend and range of dynamic stiffness variation calculated by the model proposed in this study with those from previous studies, as well as calculating the coefficient of determination (R²) between the force-displacement (or moment-angle) curves of the foot arch to characterize the similarity.”

“As shown in Table 2, for studies with stiffness results in the same units as those defined in this study, the range of foot arch stiffness is generally between 0 and 0.2 kN/mm, which is consistent with the results of this study. For studies with different units but also calculating dynamic stiffness variations, the stiffness variation trends are largely consistent when viewed from the force/displacement curves. These studies all exhibit stiffness during the heel contact phase, compliance during the plantar contact phase, a sudden increase in stiffness, and compliance during the push-off phase. The R² values from curve comparisons indicate that these results all have R² values above 0.6, with the highest reaching 0.88. These comparative results support the validity of the model proposed in this study.”

Finally, we stated the limitation of our lack of complete model accuracy validation in the end of Section Discussion: “Finally, due to the lack of practical methods for measuring foot arch loads, the model accuracy validation in this study is incomplete. This limitation arises because the proposed model is specifically designed to calculate foot arch loads and cannot provide other mechanical quantities.” and the potential future work regarding it: “Prospects also includes the potential to enhance the model by adding complexity, allowing for the calculation of other easily measurable physical quantities, thereby improving the accuracy validation method.”

3. What is the difference between calculating foot arch variation using foot arch angle changes versus using foot arch height changes?

Response 3: We appreciate your professional comments. In a more general perspective, foot arch angle changes give insight into the angular geometry and alignment of the arch, while foot arch height changes directly measure the vertical displacement, providing a more tangible understanding of how the arch behaves under different conditions [7-9]. In clinical settings, measuring arch height can be more informative for diagnosing conditions like flat feet or assessing foot func

---

## [Decision Letter · Decision Letter 1]

14 Feb 2025

A Phase Division-Based Multi-Segment Foot Model for Estimating Dynamic Foot Arch Stiffness During Walking

PONE-D-24-51122R1

Dear Prof. Liu,

We’re pleased to inform you that your manuscript has been judged scientifically suitable for publication and will be formally accepted for publication once it meets all outstanding technical requirements.

Kind regards,

Fei Yan

Academic Editor

PLOS ONE

Additional Editor Comments (optional):

Reviewers' comments:

Reviewer's Responses to Questions

**Comments to the Author**

1. If the authors have adequately addressed your comments raised in a previous round of review and you feel that this manuscript is now acceptable for publication, you may indicate that here to bypass the “Comments to the Author” section, enter your conflict of interest statement in the “Confidential to Editor” section, and submit your "Accept" recommendation.

Reviewer #1: All comments have been addressed

Reviewer #2: All comments have been addressed

2. Is the manuscript technically sound, and do the data support the conclusions?

Reviewer #1: Yes

Reviewer #2: Yes

3. Has the statistical analysis been performed appropriately and rigorously? 

Reviewer #1: Yes

Reviewer #2: Yes

4. Have the authors made all data underlying the findings in their manuscript fully available?

Reviewer #1: Yes

Reviewer #2: Yes

5. Is the manuscript presented in an intelligible fashion and written in standard English?

Reviewer #1: Yes

Reviewer #2: Yes

6. Review Comments to the Author

Reviewer #1: (No Response)

Reviewer #2: I have carefully examined the authors' responses to the reviewers' comments and the revised version of the manuscript. All the questions and concerns raised in the previous review have been addressed by the authors.

7. PLOS authors have the option to publish the peer review history of their article (what does this mean?). If published, this will include your full peer review and any attached files.

Reviewer #1: No

Reviewer #2: No

---

## [Editor Report · Acceptance letter]

PONE-D-24-51122R1

PLOS ONE

Dear Dr. Liu,

I'm pleased to inform you that your manuscript has been deemed suitable for publication in PLOS ONE. Congratulations! Your manuscript is now being handed over to our production team.

Kind regards,

on behalf of

Dr. Fei Yan

Academic Editor

PLOS ONE